# Crosstalk between Existential Phenomenological Psychotherapy and Neurological Sciences in Mood and Anxiety Disorders

**DOI:** 10.3390/biomedicines9040340

**Published:** 2021-03-27

**Authors:** Lehel Balogh, Masaru Tanaka, Nóra Török, László Vécsei, Shigeru Taguchi

**Affiliations:** 1Center for Applied Ethics and Philosophy, Hokkaido University, North 10, West 7, Kita-ku, Sapporo 060-0810, Japan; 2MTA-SZTE, Neuroscience Research Group, Semmelweis u. 6, H-6725 Szeged, Hungary; tanaka.masaru.1@med.u-szeged.hu (M.T.); toroknora85@gmail.com (N.T.); vecsei.laszlo@med.u-szeged.hu (L.V.); 3Department of Neurology, Interdisciplinary Excellence Centre, Faculty of Medicine, University of Szeged, Semmelweis u. 6, H-6725 Szeged, Hungary; 4Faculty of Humanities and Human Sciences & Center for Human Nature, Artificial Intelligence, and Neuroscience (CHAIN), Hokkaido University, Kita 12, Nishi 7, Kita-ku, Sapporo 060-0812, Japan; tag@let.hokudai.ac.jp

**Keywords:** depression, anxiety disorders, existential psychotherapy, logotherapy, meaning-centered psychotherapy, functional magnetic resonance imaging, biomarker, kynurenines, Martin Heidegger, Viktor Frankl

## Abstract

Psychotherapy is a comprehensive biological treatment modifying complex underlying cognitive, emotional, behavioral, and regulatory responses in the brain, leading patients with mental illness to a new interpretation of the sense of self and others. Psychotherapy is an art of science integrated with psychology and/or philosophy. Neurological sciences study the neurological basis of cognition, memory, and behavior as well as the impact of neurological damage and disease on these functions, and their treatment. Both psychotherapy and neurological sciences deal with the brain; nevertheless, they continue to stay polarized. Existential phenomenological psychotherapy (EPP) has been in the forefront of meaning-centered counseling for almost a century. The phenomenological approach in psychotherapy originated in the works of Martin Heidegger, Ludwig Binswanger, Medard Boss, and Viktor Frankl, and it has been committed to accounting for the existential possibilities and limitations of one’s life. EPP provides philosophically rich interpretations and empowers counseling techniques to assist mentally suffering individuals by finding meaning and purpose to life. The approach has proven to be effective in treating mood and anxiety disorders. This narrative review article demonstrates the development of EPP, the therapeutic methodology, evidence-based accounts of its curative techniques, current understanding of mood and anxiety disorders in neurological sciences, and a possible converging path to translate and integrate meaning-centered psychotherapy and neuroscience, concluding that the EPP may potentially play a synergistic role with the currently prevailing medication-based approaches for the treatment of mood and anxiety disorders.

## 1. Introduction

Mood and anxiety disorders are one of the most common diagnoses categorized in Diagnostic and Statistical Manual of Mental Disorders, 5th Edition (DSM-5) and the 11th revision of the International Statistical Classification of Diseases and Related Health Problems (ICD-11). Mental disorders account for almost 65 percent of psychiatric conditions worldwide and nearly 30 percent of the population is affected by mental illness during their lifetime [1]. A selective serotonin reuptake inhibitor (SSRI) fluoxetine is the most effective pharmacotherapy for the acute treatment of moderate-to-severe depressive disorder in children and adolescents [2]. However, only 40–60% of individuals relieved their symptoms with antidepressants within six to eight weeks [3]. Furthermore, intervention for treatment-resistant depression (TRD) and prevention of suicide attempt are a major challenge. Mood and anxiety disorders elevate morbidity and mortality, present a high rate of comorbidity with medical conditions, and impose a great social burden. The comorbidity complicates symptoms, limits choice of pharmacological treatments, lowers treatment adherence, increases the use of healthcare service, and thus increases costs [4]. Treatment options for psychological symptoms are even more limited for post-myocardial infarction patients and patients in terminal care due to advanced cancer [5,6].

A meta-analysis reported once that pharmacotherapy with SSRIs was significantly more effective than psychotherapy in patients with major depressive disorder [7]. However, a network analysis revealed that combined pharmacotherapy and psychotherapy was superior to monotherapy, while no significant difference was found between pharmacotherapy and psychotherapy in the treatment of depression [8]. An increasing number of therapists pay more attention to psychotherapy as an effective adjunct in interventions for psychiatric disorders. Psychotherapy has presented successful clinical outcomes, which are consistent with less than five percent variance among skillful psychotherapists. Psychotherapy has also given deeper insights into human behavior, but it still stands in need of empirical assessments and methodologies [8]. It would undoubtedly be useful to investigate in what way and how psychotherapy modifies the complex brain responses underlying mental illness, which may lead to the development of new therapeutic interventions. Existential phenomenological psychotherapy (EPP) has been at the forefront of meaning-centered psychotherapy (MCP). Meanwhile, the progress in neuroimaging research such as functional magnetic resonance imaging (fMRI) techniques has provided more and more knowledge of brain functions, which leads to deeper understanding of psychopathologies and the development of therapeutic interventions for mental disorders [9]. Neuroscience in vitro and in animal models have allowed us to test hypotheses, reveal pathogenesis, and develop new drugs for psychiatric disorders [10]. Therefore, translating and integrating methods and knowledge of psychotherapy and neurological sciences can certainly lead to deeper understanding of disease mechanisms, testing new hypotheses, and searching for novel treatments of mental illness [11]. This review article presents the development and methods of EPP, clinical evidence of its efficacy, current understanding of mood and anxiety disorders in neurosciences, and the need for convergence of this expertise to translate and integrate MCP and neurological sciences (Figure 1).

## 2. Existential Phenomenological Psychotherapy

Phenomenology as a philosophical approach of grasping and explaining reality appeared on the European intellectual scene some 130 years ago. Although G.W.F. Hegel already talked and wrote extensively about the “phenomenology of spirit” in the early 19th century, it was not until Edmund Husserl’s adaptation of the term in the early 1900s that it came to gain its philosophically characteristic, nuanced meaning. Husserl, a mathematician and philosopher, was strongly influenced in the construction of his methodology by philosopher and psychologist Franz Brentano and by historian and hermeneutic philosopher Wilhelm Dilthey. From Brentano he took over arguments regarding the structure of consciousness as well as the general task of critiquing psychologism in the domain of philosophical logic, whereas from Dilthey he appropriated the “attempt to develop a more descriptive approach to the human sciences” [12]. Having carefully designed his innovative philosophical methodology, which he believed “could discover the structures common to all mental acts”, he had high hopes that the systematic utilization of this methodology could turn philosophy into a “rigorous science” on par with the natural sciences. Thus, was born phenomenology, a novel philosophical approach whose concise programmatic ideal was aptly conveyed in the following famous slogan: “back to the things themselves”.

What does Husserl mean by “back to the things themselves”? This motto involves a call, an appeal to turn away and leave behind concepts and theories, and instead to experientially encounter all phenomena of the pure consciousness just as they present themselves, without the conceptual burden with which language and tradition weighs them down. An eminent historian of the phenomenological movement, Herbert Spiegelberg, noted that transcendental phenomenology attempted to reveal “the structures of pure consciousness, [which were] made accessible by a special suspension of belief in the reality of our natural and scientific world, the so-called phenomenological reduction, in which the constitution of the phenomena according to intending acts and intended contents was studied in detail” [13]. This phenomenological reduction is essential to understanding how Husserl’s phenomenology was intended to operate. As Husserl explained, the main objective of phenomenology’s undertaking was to intuitively discern eidetic essences that lay beyond the singular expressions of individual phenomena [14]. While descriptive phenomenology’s primary goal was to “intuit, analyze, and describe the data of direct experience in a fresh and systematic manner”, essential or eidetic phenomenology’s responsibility was to explore “essential structures on the basis of imaginative variation of the data” [13].

The phenomenological approach had progressively become influential in psychological and psychiatric circles from the second third of the last century; especially after it successfully wed itself with the newly emerging philosophical and ethical concerns of existentialism. Martin Heidegger, who was originally an assistant of Husserl, and whose philosophical reputation by the 1930s came to match his distinguished former teacher’s considerable intellectual stature, was the one who, albeit unwittingly and increasingly unwillingly, facilitated this fruitful philosophical nuptial. In his famous 1927 work ‘Being and Time’, Heidegger critically transformed Husserl’s insights and carried them over to the spheres of ontology [13]. Ontology as a specialized philosophical field deals with the existence—or being—of all beings, and tries to find answers to such questions as what “being” means, what separates being from non-being, what levels of existence and various classes of entities there are, etc. Heidegger’s quest for the forgotten meaning of “being”, and particularly the “existential analytic” of Being and Time that expounded on the self’s—or as Heidegger had called it, Dasein’s—existential possibilities and limitations, initiated a prominent philosophical and literary movement that came to be called “existentialism”. Among the proponents of existentialism influential figures customarily counted are Jean-Paul Sartre, Albert Camus, Simone de Beauvoir, Karl Jaspers, Maurice Merleau-Ponty, and so forth. Nevertheless, the rapidly growing existentialist movement of the 1940s and 1950s also began to claim an outstanding philosophical ancestry that ranged from the works of Arthur Schopenhauer to Soren Kierkegaard and from Fyodor Dostoevsky to Friedrich Nietzsche. What was common in this exceptionally diverse bunch of thinkers was the circumstance that they were all deeply intrigued by the “subjective” experiences and the inwardness of the self, at the same time looking distrustfully upon the claims for scientific hegemony of the positivist, reductionist, and objectifying approaches of the natural sciences (Figure 2a).

Discovering the meaning of human existence was the first and foremost task which the existentialist thinkers set for themselves. As Wrathall and Dreyfus eloquently put it: “For existentialist thinkers, the focus is on uncovering what is unique to that individual, rather than treating her as a manifestation of a general type (…) With their focus on the individual and a denial of any meaningful sense of what constitutes an essential or absolute goal for human existence, existentialists emphasize human freedom and responsibility, and hold that the only goal consistent with that freedom and responsibility is to live authentically” [12]. When phenomenology began to merge with existentialism, first in Heidegger’s, then in a prestigious line of other up-and-coming philosophers’ work, the rigorousness of phenomenology and the existential concerns of existentialism generated a unique blend of philosophizing. As Spiegelberg rightly points out, this new combined approach was in palpable contrast with the methodologically less scrupulous early existentialist writings of for instance Kierkegaard or Nietzsche. The new phenomenological-existential approach firmly maintained “that existence *can* be approached phenomenologically and studied as one phenomenon among others in its essential structures” [12]. Wrathall and Dreyfus add to this that since phenomenology and existentialism share many of their concerns, their joint focus tends to be on these concerns, rather than on numerous other issues that they would disagree on. “Like the phenomenology of Heidegger, Merleau-Ponty, and Sartre, existentialism as a movement starts its analysis with the existing individual—the individual engaged in a particular world with a characteristic form of life. Thus, an emphasis on the body and on the affective rather than rational side of human being are characteristic of existentialism” (Figure 2b) [12].

This is the tradition upon which existential psychiatry and psychotherapy has established its theory and practice. Existential psychotherapy is a unique approach in that it scrutinizes psychic phenomena from an existentialist point of view. This entails that existential therapists understand man not as a substantive subject that is torn from objective, “external” reality, but as a Being-in-the-World (Heidegger’s term) or existence. Existential psychiatry began forming shortly after the first reception of Heidegger’s Being and Time—that is, in the early 1930s—, but it became influential only from the 1950s and 1960s. In the beginning, such psychiatrists as Ludwig Binswanger and Medard Boss had attempted to elaborate the putting into practice of what Heidegger had to say about Dasein’s fundamental ontological structures [15]. Under the name of ‘Daseinsanalysis’, both Binswanger and Boss had worked out a theory as to how one might utilize phenomenology’s insights in understanding and curing the mentally ill. Around the same time Austrian-born Viktor Frankl also established a distinct existential school of psychiatry and psychotherapy for which he coined the name Logotherapy (Figure 3a).

It is important to keep in mind that existential-phenomenological psychiatry and psychotherapy were born out of the elemental dissatisfaction with the ways psychiatrists tried to cure conventionally conceived “diseases” of conventionally conceived human “subjects”: for existentialists it seemed that neither of these were adequate modes of either grasping human reality or of trying to advance the well-being of the individual [16]. By the end of the 1950s, the existential approach to psychiatry and psychotherapy arrived and started to grow in the US. Along with the so-called humanistic movement in psychology, it has gained considerable influence after a relatively brief period of time. Rollo May and Irvin D. Yalom are the most well-known and acknowledged representatives of the American school of existential therapy. Meanwhile in Great Britain, it was Ronald David Laing who first offered analyses of psychopathological and psychotherapeutic phenomena from an existential-phenomenological stance. Somewhat later, several members of the British School of Existential Analysis came to reform the principles of existential therapy. Their objective was to strip the therapeutic setting from the urge to diagnose the “disease” of the client, along with the removal of the habitual pathologization of various “abnormal” psychological phenomena and the moral evaluation of medical data. Instead, they suggested that therapy should focus on the phenomenological description of the “lived world” of the patient. Ernesto Spinelli, leading representative of the British School of Existential Analysis, asserted that existential psychotherapy’s “primary task is *not* one of seeking to direct change in the worldview of the client. Rather, existential psychotherapy’s principal concerns lie with its attempts to descriptively clarify that worldview so that its explicit and implicit, sedimented dispositional stances can be re-examined inter-relationally” [17] (Figure 3b).

### 2.1. Existential Psychotherapy’s Approach for the Treatment of Mood and Anxiety Disorders

The aim of existential analysis is to guide a person towards experiencing their life authentically and freely. This is done through practical methods that help an individual to live with ‘inner consent’, or the ability to affirm what he or she is doing [18].

As Viktor Frankl, the founder of the Vienna School of Existential Analysis maintained, existential analysis is the search for meaning: the meaning of a given individual’s unique existence with its goals and values, amidst its limitations and situatedness [19]. However, this approach to the human psyche comprises other essential facets as well that point beyond a mere psychological analysis of contingent factors. Frankl claimed that “existential analysis is (…) not only the explication of ontic existence, but also the ontological explication of what existence is. In this sense, existential analysis is the attempt at a psychotherapeutic anthropology, an anthropology that precedes all psychotherapy, not only logotherapy.” The establishment of such an anthropology is the ultimate goal of existential psychology and psychiatry as a theory of the human psyche and its dysfunctions. It is exceedingly important to clarify such an anthropology, because, as Frankl goes on to explain: “every psychotherapy plays itself out against an a priori horizon. There is always an anthropological conception at its foundation, no matter how little aware of this the psychotherapy may be” [19].

Frankl, an Austrian-born Holocaust survivor, believed that the root cause of the majority of our psychological problems was a general feeling of meaninglessness: an “existential vacuum” which leads one to despair and believe that life no longer has any meaning [19]. Therefore, the goal of his therapeutic approach is to assist rediscovering meaning in one’s life (logotherapy: meaning therapy). One might wonder why meaning appears crucial for psychiatry and psychopathology. The answer lies in the fact that mental health and the perceived meaningfulness of personal existence seem to be both coemerging and codependent. Numerous studies have demonstrated that the absence of a comprehensive framework of meaning which includes goals, values, and priorities in an individual’s life is strongly correlated with the formation of depressive disorders [20]. On the other hand, the presence of meaning has been shown to be an active and potent protective factor against the emergence of suicidal tendencies which are among the most dangerous potential consequences of depression [21,22].

Besides depression, other psychological illnesses such as anxiety disorders can also be directly linked with a lack of a sense of overall meaning in life. “Although skeptical scholars criticize meaning in life as a tenuous construct, research shows that many individuals perceive there to be a larger direction and orientation in their daily lives, and when they lack this experience, they seem more prone to developing depression, anxiety, and other psychological problems” [23].

Regarding some of the concrete techniques that EPP routinely employs to treat mental disorders, the following two are of special interest: paradoxical intention and dereflection [19] (Table 1). Frankl argued that both with anxiety disorders and with obsessive-compulsive disorders, the technique of paradoxical intention can be effectively utilized. As he wrote, “we define paradoxical intention in the following way: the patient will be directed to wish (in the case of anxiety neuroses) or to resolve to do (in the case of compulsive neuroses) precisely that which the patient fears so much” [19]. In other words, paradoxical intention is a technique whereby the individual resolves themselves to opt for an attitude that is diametrically opposed to that which they would originally want to adopt as a “natural” reaction to their perceived psychological difficulty. For example, if one is pathologically anxious and is terrified of having a panic attack when speaking in front of an audience (glossophobia), then Frankl would suggest that instead of trying to avoid the anxiety-inducing thoughts, on the contrary, they should engage these troubling thoughts head-on and even exaggerate them. As Vos elucidates, “paradoxical intentions are based on the assumption that individuals can choose the stance they take towards their psychological difficulties and that their symptoms are exacerbated by avoiding problems or feeling saddened or anxious. Frankl invited clients to deliberately practice or exaggerate a neurotic habit or thought, so that they stopped fighting and instead identified and undermined their problems. This technique has proven to be particularly effective in anxiety disorders [23] (Table 1).

The other technique of EPP is called dereflection. It follows the opposite route of paradoxical intention: instead of directly engaging the issues with which the individual is principally preoccupied, the attention gets redirected, away from the self, towards other people or other phenomena in the world. As a rule, this technique is used when the client becomes overly self-absorbed with their own goals and problems. The excessive absorption with one’s own problems is what Frankl called hyperreflection. In the following passage he refers to the example of sexual impotence which is caused by psychological dysfunction. “In logotherapy we counter hyperreflection with a dereflection. To treat the specific hyperintention that is so pathological in cases of impotence we have developed a special technique, which dates back to 1947. We recommend that the patient be encouraged ‘not to engage in sex, but rather to acquiesce to fragmentary acts of tenderness, like a mutual sexual foreplay” [19]. Consequently, by the drawing away of the attention from the perceived problem—sexual impotence—and thus from the self towards the other person—by giving tenderness and caring, as well as by other gentle forms of mutual pleasuring—the psychological block can be gradually lifted, and the sexual functioning can become normal once again (Table 1).

### 2.2. Clinical Evidence of Meaning-Centered Psychotherapy

The EPP is practiced as a MCP in general. MCP has proved its efficacy against depressive and anxiety symptoms in patients with a wide range of diseases from psychiatric disorders, cardiovascular disease to terminal cancer. Systematic search was conducted in PubMed/Medline with keywords “meaning-centered” and “psychotherapy” on 16 December 2020. Fifty-three articles were retrieved, and twelve articles were eventually deemed appropriate for synthesis (Appendix A).

An exploratory pilot study reported that advanced cancer patients receiving home palliative care showed a significant decrease in levels of despair, anxiety, depression, and emotional distress by receiving individual MCP (IMCP), compared to those who received only counseling [24]. Randomized control trials showed the efficacy of IMCP and meaning-centered group psychotherapy (MCGP) for psychological and existential distress in patients with advanced cancer. The IMCP and MCGP were superior to enhanced usual care and supportive psychotherapy [25,26,27,28]. In MCGP the improvements of quality of life, depression, hopelessness, and desire for hastened death in advanced cancer patients were mediated by an enhanced sense of meaning and peace in life [29]. A longitudinal mixed-effects model also showed significant increases in alleviating burden, anxiety, and depression and finding meaning, benefit, and spiritual well-being among cancer caregivers in response to web-based MCP [30]. A one-year follow-up study showed that cancer survivors who completed MCGP presented more personal growth than those who received supportive group psychotherapy. A two-year follow-up study reported that MCGP cancer survivors showed better positive relations than usual care receivers, suggesting that MCGP carries higher efficacy in the long term [31]. The effectiveness and cost-effectiveness measurements of MCGP for cancer survivors have been designed to compare meaning making, quality of life, anxiety and depression, hopelessness, optimism, adjustment to cancer, and costs with supportive group psychotherapy and usual care [32].

The results may reinforce the evidence of the efficacy and determine the cost-benefit ratio of MCGP. Furthermore, an open trial study reported the preliminary results that meaning-centered grief therapy for parents who lost a child to cancer presented improvements in prolonged grief, sense of meaning, depression, hopelessness, continuing bonds with their child, posttraumatic growth, positive affect, and quality of life. The treatment gains were maintained or improved after three months [33]. The meaning-centered intervention was shown to provide perceived benefits to palliative care nurses who faced recurrent burden, but improvement of spiritual and emotional quality of life remains unclear [34].

A meta-analysis which included 60 trials and 3713 samples reported that MCP had large effect sizes on quality of life and psychological stress in the immediate time frame and follow-up compared to controls. Quality of life is larger in effect size in the immediate time frame than meaning in life, hope and optimism, self-efficacy, and social well-being. Moreover, meta-regression analysis revealed that meaning in life is a predictor of psychological stress [35].

In summary, both IMCP and MCGP are more effective than supportive psychotherapy, counselling, or supportive care. MCGP nurtures personal growth and positive relations. MCGP is effective in the long term and more cost-effective. However, most MCP studies mentioned above dealt with terminal cancer patients presenting mood symptoms. MCP analysis targeting a population of neurologic or psychiatric diseases, MCP for individuals without comorbidity, and comparison with patients under pharmacotherapy will further reveal the efficacy, the applicability, and the limits of MCP for the treatment of a wide range of diseases.

## 3. Neurological Sciences’ Approach to Mood and Anxiety Disorders

### 3.1. Neuroimaging

Recent advances in neuroimaging technology have facilitated the investigation of brain structure and function. Among magnetic resonance imaging, computed tomography, and positron emission tomography, functional MRI (fMRI) provides information on the properties of functional connectivity (FC). Resting-state fMRI investigates behavioral characteristics such as psychological states, sustained attention, personality, temperament traits, creative ability, and cognitive ability including working memory and motor performance [36,37,38,39]. Furthermore, the patterns of resting-state fMRI are correlated with specific symptoms and respond to treatment [40,41]. Analytical methods of resting-state network connectivity include seed-based analysis, the amplitude of low-frequency fluctuation (ALFF) and fractional ALFF techniques, regional homogeneity (ReHo), independent component analysis (ICA), and graph theory.

#### 3.1.1. Functional Magnetic Resonance Imaging

##### The Default Mode Network

The default mode network (DMN) is a network of interacting brain regions which shows synchronized activation and deactivation during tasks [42]. DMN includes the medial prefrontal cortices (mPFC), the posterior cingulate cortex (PCC), precuneus, inferior parietal lobule, lateral temporal cortex, and hippocampal formation [43,44]. DMN activity is associated with internal processes including self-referential thinking, autobiographical memory, or thinking about the future [45,46,47]. The DMN is divided into an anterior subdivision centered in the mPFC and a posterior subdivision centered in the PCC. The anterior DMN is more related to self-referential processing, and emotion regulation through its strong connections with limbic areas. The posterior DMN is associated with consciousness and memory processing through its connection with hippocampal formation [48,49] (Figure 4).

A relative increase in DMN connectivity and significant ReHo reduction were observed in the posterior DMN of patients with late-life depression (LLD) [50,51,52,53]. ICA studies revealed an increased connectivity within the anterior DMN of patients with depression compared to healthy controls [54]. The dissociation between the anterior and posterior DMN subdivisions was observed in patients with major depressive disorder [55]. Antidepressant treatment restored FC abnormality in the posterior DMN but did not correct the FC abnormality in the anterior DMN. Network homogeneity was increased in the anterior DMN but decreased in the posterior DMN [56]. Seed-based analysis using mPFC showed the dissociation between the anterior and posterior DMN and increased connectivity between the anterior DMN and the salience network (SN) in depression [57,58]. Decreased PCC connectivity and increased connectivity in the anterior DMN were observed in depressive patients without medication and 12-weeks treatment of paroxetine partially restored the decreased connectivity [59]. In general, FC between PCC and left medial frontal gyrus decreased in patients with depression and 12-weeks of antidepressant treatment increased FC between PCC to the bilateral medial frontal gyrus [60]. Psychedelics are known to disrupt the activity of the DMN. Serotonergic psychedelic psilocybin-assisted therapy significantly reduced the depression scores of patients with severe depression [61].

##### The Executive Control Network

The executive control network (ECN) plays an important role in the integration of sensory and memory information, the regulation of cognition and behavior, and the process of working memory [62]. The ECN consists of the dorsolateral prefrontal cortex (dlPFC), medial frontal cortex, lateral parietal cortex, cerebellum, and supplementary motor area [63]. Changes in the ECN were reported in ageing and in patients with LLD, mild cognitive impairment, Alzheimer’s disease, and Parkinson’s disease [64,65,66,67,68] (Figure 4).

Disruptions of the ECN were reported in non-demented elders with LLD [69,70,71]. Seed-based analyses using the dlPFC showed decreased FC in the frontoparietal areas in patients with LLD and current depression [72]. Seed-based analyses of the cerebellum presented decreased FC in ECN nodes in dlPFC and the parietal cortex and DMN nodes [73,74]. ICA analyses reported decreased FC in the dlPFC and superior frontal areas, which is consistent with other resting-state fMRI studies with ReHo and ALFF [52,75,76,77]. Decreased FC in the frontal-parietal cortex was also reported in LLD remitters 3 months after remission [78]. Alteration of the ECN was associated with susceptibility to distraction, and difficulty in sustaining attention, multi-tasking, organizational skills, and concrete thinking [79]. The FC between the dlPFC and other bilateral regions was negatively associated with executive function in patients with LLD [80]. Furthermore, the levels of functional disability were positively correlated with executive dysfunction in LLD [81,82]. Low and slow response to antidepressants and relapse were correlated with deficits in word-list generation and response inhibition which are governed by the executive function network [83,84]. In addition, dissociation between the posterior DMN and ECN was also reported in patients with LLD and current depression [85,86].

##### The Salience Network

The SN detects and filters salient stimuli and recruits relevant functional networks [87]. The SN is responsible for detecting and incorporating sensory and emotional stimuli, allocating attention, and switching inward and outward cognition. The SN is located in the ventral anterior insula and includes nodes in the amygdala, hypothalamus, ventral striatum, and thalamus [88]. The ventral components play a role in emotional control, while the dorsal components play a role in cognitive control [89]. Cognitive tasks activate the dorsal components including the dorsal anterior cingulate cortex and the right anterior insula. During cognitive tasks, the SN engages ECN and disengages DMN, but vice versa in rest [89,90,91]. Dissociation between the ECN and SN is correlated with cognitive task performance [92] (Figure 4).

Decreased FC from the amygdala to the hippocampus was observed in patients with depression and individuals at high risk of depression [93,94]. A disrupted pattern of SN connectivity was reported in depression, especially in the insula and amygdala [95]. Elevated connectivity was found between the insula and DMN in patients with LLD [96]. Seed-based analysis using the amygdala as a seed region was positively associated with increased amygdala FC with DMN nodes and long-term negative emotions [97]. Increased FC between the SN and DMN is considered to predispose individuals to depression but decreased FC between the amygdala and precuneus was reported in patients with depression [70,98,99]. Decreased negative FC between the ECN and the SN was associated with cognitive impairment and severity of depression in patients with LLD. Disrupted standard SN pattern was associated with a worse treatment response [100].

#### 3.1.2. Task-Related Functional Magnetic Resonance Imaging

Mood disorders, anxiety disorders, and posttraumatic stress disorder (PTSD) share neurobiologically common characteristics in task-related fMRI. A meta-analysis was conducted using articles studying stereotactic coordinates of whole-brain-based activation in task-related fMRI as between adult patients and controls [101]. Patients with mood disorders, anxiety disorders, or PTSD shared abnormalities in convergence of task-related brain activity in regions associated with inhibitory control and salience processing [101]. Patients who suffered from mood and anxiety disorders presented abnormally lower activity in the inferior prefrontal and parietal cortex, the insula, and the putamen [101]. These regions are responsible for cognitive and motional control, and inhibition of and switching to new mental activities. The patients also showed higher activity in the anterior cingulate cortex, the left amygdala, and the thalamus which process emotional thoughts and feelings [101].

### 3.2. Other Relevant Biomarkers and Therapeutic Targets

Besides the large-scale brain network, natural products, endogenous metabolites, neuropeptides, receptor agonists, their synthetic analogues, plasma proteins, and lipids are under extensive study in search of biomarkers and novel drugs for mental disorders [102,103,104,105,106,107,108,109]. In addition, the disruption of neural circuitry-neurogenesis coupling was observed in depression [109]. Several neurotransmitters including serotonin, dopamine, adrenaline, histamine, gamma-aminobutyric acid, and peptides play an important role in the pathogenesis of mood and anxiety disorders. Selective serotonin reuptake inhibitors (SSRIs), selective norepinephrine reuptake inhibitors (SNRIs), and monoamine oxidase inhibitors (MAOIs) are major classes of antidepressants currently prescribed for the treatment of depression and anxiety. SSRIs, SNRIs, and MAORs all act on components of neurotransmission. Serotonergic psychedelics are a subclass of hallucinogens that act on the serotonin 5-HT_2A_ receptors. The naturally occurring psychedelic prodrug psylocibin was reported to alleviate depression and anxiety in patients with life-threating diseases [106]. Glutamatergic neural transmission is drawing increasing attention because normal human brain functions are maintained in balance of 80% of excitatory neuronal and 20% of inhibitory neuronal activities [110]. Excitatory neurotransmission is governed by glutamatergic neurons with the N-methyl-D-aspartate (NMDA) receptor [111]. NMDA receptor antagonists are under extensive study for the treatment of TRD [112]. The subanesthetic dose of NMDA receptor antagonist ketamine rapidly improves depressive symptoms and leads to the resolution of suicidal ideation in patients with serious depression [113]. However, the NMDA receptor appears not to be a single pharmacological target of ketamine in the alleviation of depression [114].

Kynurenines (KYNs) are intermediate metabolites of the tryptophan (TRP)-KYN metabolic pathway, which exhibit a wide range of bioactivity such as neurotoxic, neuroprotective, oxidative, antioxidative, and/or immunological actions [115]. The KYN metabolites include a NMDA receptor agonist as well as a NMDA antagonist [116]. Furthermore, the KYN pathway supplies neuroactive metabolites which trigger biological functions not only in synaptic spaces, but also in the non-synaptic microenvironment around the neurons [117]. Moreover, increasing attention has been paid to the KYN pathway since over 95 percent of TRP is metabolized through the KYN pathway, leaving about one percent to the synthesis of serotonin that plays an important role in mood disorders. Kynurenic acid (KYNA) is found to be a diagnostic as well as predicative biomarker for depression, while KYN and KYNA are potential predictive biomarkers for escitalopram treatment in depression [118]. KYNs are agonists or antagonists at the NMDA receptor of the glutamatergic nervous system. Thus, the glutamatergic nervous system has been proposed to be a target for mood disorders [110]. A meta-analysis concluded that an increased risk of depression was correlated with inflammation in chronic illness through the TRP-KYN metabolic pathway [119]. A systematic review reported KYN metabolism abnormalities in TRD and suicidal behavior, proposing the KYN enzymes as novel targets in TRD and suicidality [120].

Gastrointestinal microbiota were observed to participate in development of visceral pain, anxiety, depression, cognitive disturbance, and social behavior and microbiota composition was proposed to be a potential biomarker and target [121,122]. Serum plasma profiles may serve as a potential predictive biomarker for the choice of antidepressants [123]. Foods, or fortified food products beneficial to physiological body functions, were proposed for the treatment of metabolic dysfunction in ageing neurodegenerative diseases [124]. In addition to biomolecules, any measurable indicators are important for risk, diagnosis, prognostic, and predictive biomarkers and interventional targets. Depression was found to be a risk factor for Alzheimer’s disease and dementia. Dyslipidemia treatment reduced the risk of development of dementia in diabetics [125]. The presence of depressive symptoms following acute stroke or transient ischemic attack increased mortality and disability within the following 12-month period, suggesting that depression is a prognostic biomarker in cerebral ischemia [126]. Therefore, the treatment of depression is a crucial measure to avoid the development of comorbid conditions and psychotherapy is certainly able to contribute to the prevention of disease progression and complications for a better quality of life. In addition, depression is a measurable psychobehavioral component of dementia, which can be ameliorated by animal-assisted and pet-robot interventions in patients with dementia [127] (Figure 5).

## 4. Future Perspective: Bridging the Expertise

Clinical evidence including randomized-controlled clinical trials and meta-analysis presented solid evidence that EPP, which is generally practiced as MCP, provides substantial relief to individuals under psychological stress such as depression and anxiety. MCP improves quality of life and ameliorates psychological stress, while meaning in life predicts the degree of psychological stress in patients with advanced cancer.

Neurological sciences have shed more light on the understanding of the pathomechanisms of depression and anxiety and explores possible interventional targets. Imaging studies and various network analyses have provided enormous data on and presented possible mechanisms of mood and anxiety disorders, but changes in the networks in individuals under psychotherapy have been rarely reported. Neither have been measurable biomarkers in individuals under psychotherapy. Recent research revealed that depression itself is a risk, diagnostic, prognostic, and/or predictive biomarker for various diseases such Alzheimer’s disease, dementia, strokes, major depressive disorders, and chronic diseases (Figure 5). Thus, depression may be a desirable interventional target to reduce the risk, morbidity, disability, and mortality of illness. It has been observed that neurologic diseases are commonly associated with psychiatric comorbidities, exposing neurologic patients to a sense of hopelessness and consequently a higher risk of suicidal ideation and suicidal behavior. Thus, neurologic diseases may be the ideal territory for bridging existential psychotherapy and neurological sciences from the side of clinical medicine [21,22]. Meanwhile, characterizing the behavior traits of animal models of neurologic and psychiatric diseases and developing animal models of happiness and wellbeing to assess hedonic and eudaimonic components, if there are any, may pave an approach toward the converging point from the side of laboratory medicine. On the other hand, the introduction of DSM-5 and ICD-11 has accelerated a trend toward discounting the psychological and social factors of psychiatric disorders. The roles of transdisciplinary psychiatry are expected to be explored and refined to fill the gap between psychiatry and neuroscience [128].

Further translational and integrative research on both psychotherapy and neuroscience is expected in order to provide symptomatic relief from psychological stress, to prevent the development of comorbidity, and to avoid exacerbation of diseases, especially for individuals contraindicated to pharmacotherapy, making available more options for treatment and realizing a possible individualized combination therapy based on psychotherapy and pharmacotherapy.

## Figures and Tables

**Figure 1 biomedicines-09-00340-f001:**
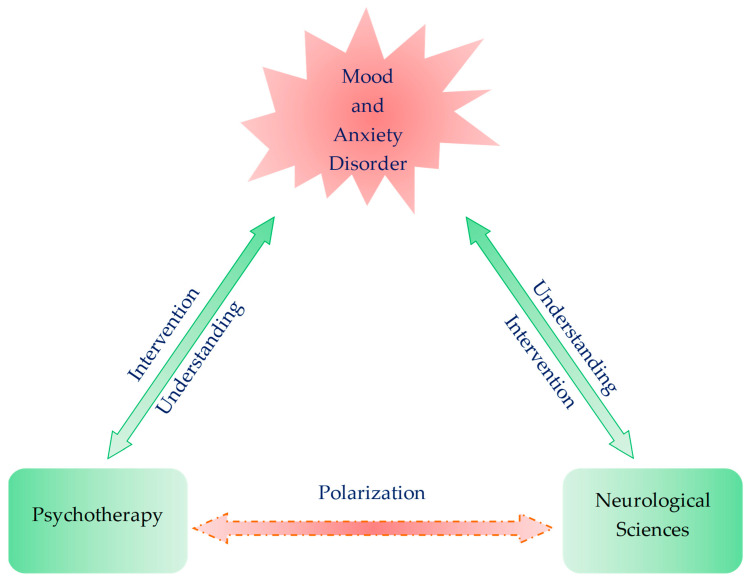
Polarization of two fields of expertise: psychotherapy and neurological sciences. Both psychotherapy and neurological sciences explore the pathomechanism and interventional opportunity of psychiatric disorders. Translational studies are scarce, and each area of expertise stays polarized.

**Figure 2 biomedicines-09-00340-f002:**
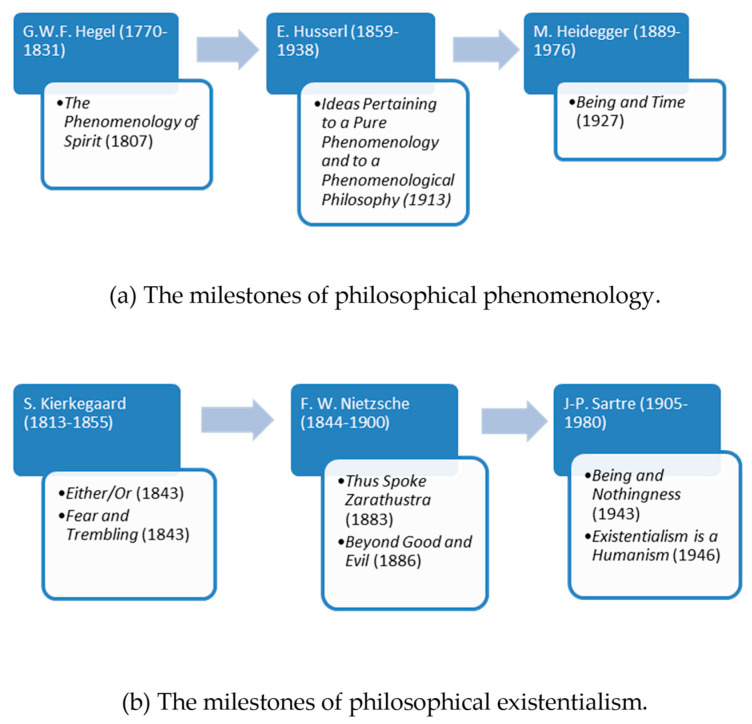
The milestones of philosophical phenomenology and philosophical existentialism. (**a**) The milestones of philosophical phenomenology. (**b**) The milestones of philosophical existentialism.

**Figure 3 biomedicines-09-00340-f003:**
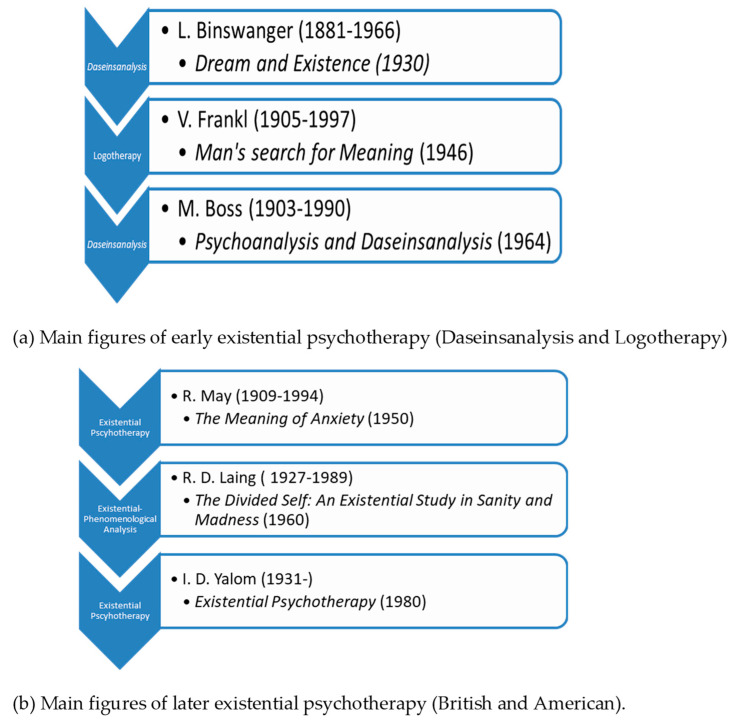
Main figures of early existential psychotherapy and later existential psychotherapy. (**a**) Main figures of early existential psychotherapy (Daseinsanalysis and Logotherapy). (**b**) Main figures of later existential psychotherapy (British and American).

**Figure 4 biomedicines-09-00340-f004:**
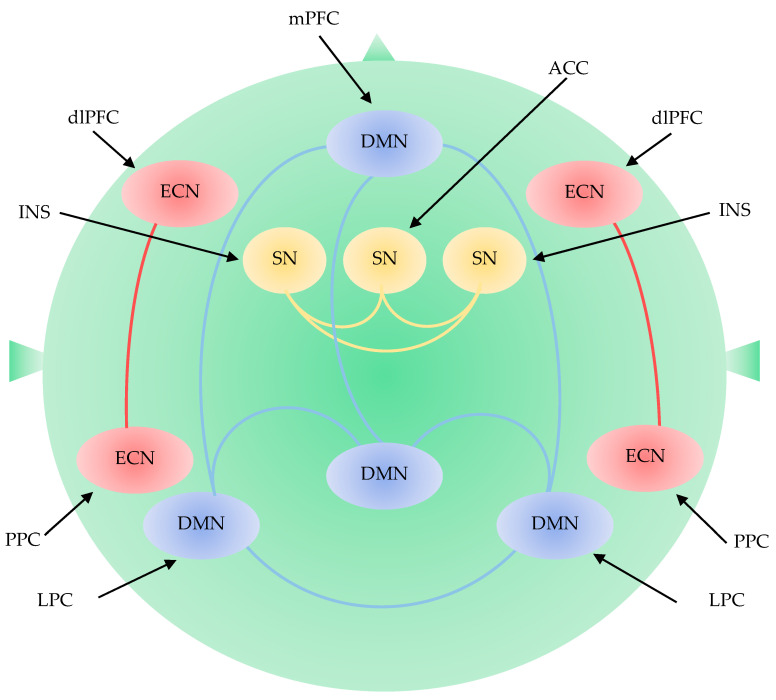
Large-scale brain network including the default mode network, the executive control network, and the salience network. DMN: default mode network; ECN: executive control network; SN: salience network; ACC: anterior cingulate cortex; dlPFC: dorsolateral prefrontal cortex; INS: insular cortex; LPC: lateral parietal cortex; mPFC: medial prefrontal cortex; PPC: posterior parietal cortex.

**Figure 5 biomedicines-09-00340-f005:**
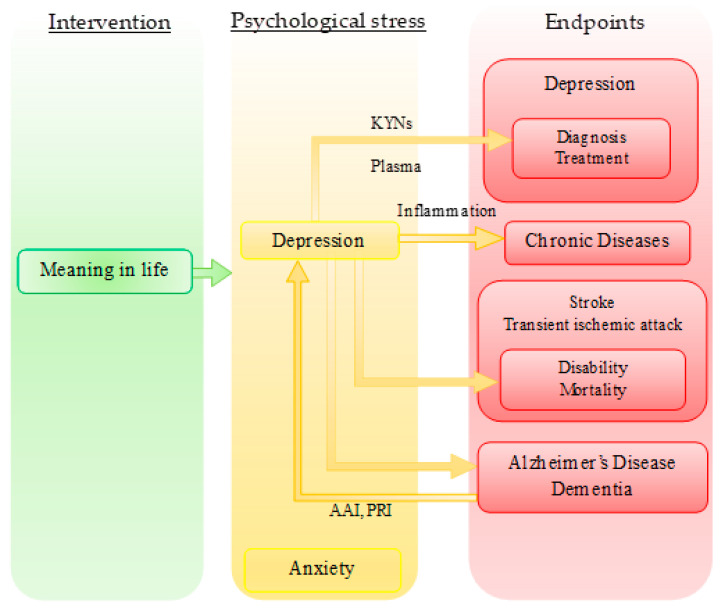
Meaning-centered psychotherapy, its effective targets, and endpoints. Meaning in life is a predictor of psychological stress. Psychological stress causes depression, anxiety, and cognitive impairment. Depression is a measurable indicator which predicts diagnosis and/or treatment of depression with kynurenines (KYNs), chronic diseases with inflammation, disability and mortality of stroke and transient ischemic attack, and Alzheimer’s disease and dementia. Depression of Alzheimer’s diseases and dementia can be ameliorated by AAI (animal-assisted intervention) and pet-robot intervention (PRI).

**Table 1 biomedicines-09-00340-t001:** Techniques of existential psychotherapy.

Name of the Technique	How Does It Work?	Application
Paradoxical intention	Resolves to opt for an attitude that is diametrically opposed to that which they would originally want to adopt as a “natural” reaction to perceived psychological difficulty	Anxiety disorders Depression [18].
Dereflection	Redirect the attention from the self, towards other people or other phenomena in the world	Anxiety disorders Depression [18].

## Data Availability

The data that support the findings of this study are available from the corresponding author, L.B., upon reasonable request.

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
