# Peer review of "Crosstalk between Existential Phenomenological Psychotherapy and Neurological Sciences in Mood and Anxiety Disorders"

_biomedicines, 2021, doi:10.3390/biomedicines9040340_

Round 1
Reviewer 1 Report
Thank you for the opportunity to review this extremely interesting text. It addresses in an original way an issue of increasingly emerging interest and helps to fill the need for an idispensable dialogue between psychotherapy and neuroscience. It is also very well presented with informative and graphically pleasing diagrams.
I would like to highlight just a few points, which the authors are encouraged to address:
- In the abstract (lines 32-33) and in the initial part of the introduction (lines 47-48): I believe it is appropriate to soften the expressions describing psychotherapy as a "potent alternative to the currently prevailing medication-based approaches" and "a preferable otion for intervention of psychiatric disorders." Surely the psychotherapy is a very powerful mean, but I fear that the sentences in object could be generalized and misunderstood. In order not to go into the specific guidelines that, depending on the different disorders, distinguish the indications for psychotherapy versus psychopharmacological treatments, I recommend an expression suggesting "a synergistic action in the vast majority of cases" or a similar expression.
- Paragraph 3, lines 207-209. This sentence is clearly crucial in the text. I believe it deserves to be enriched by the mention of one of the most dangerous consequences of depression, suicide (just the mention, a more extensive elaboration is not necessary in the economy and purpose of the text). This, also in terms of existential perspective. In fact, the presence of meaning has been shown to be a powerful protective factor against suicide. I suggest two texts, which the authors can read and cite: 1) “The meaning in life in suicidal patients: The presence and the search for constructs. A systematic review”. Medicina (2019), 55(8), 465 (open access full PDF text); 2) “Suicide prevention from a public health perspective. What makes life meaningful? The opinion of some suicidal patients”. Acta Biomed (2020), 91(3-S), 128-134 (full PDF text available on Google Scholar). In the first text there is a historical introduction to the evolution of the concept of meaning and a review of that concept in light of the phenomenon of suicide. The second is instead a qualitative study, which insists - as the authors rightly reiterate - on the subjective experience of meaning in suicidal patients. Both texts advocate psychotherapeutic treatment focused on meaning, following in the footsteps of V. Frankl's logotherapy.
- Paragraph 5. This is also an interesting and well-written paragraph. I agree with the authors who highlight the possible role of neuroinflammation. Again, important scientific evidence has been demonstrated in the context of suicide (look, for example, at this work: “Abnormalities in Kynurenine Pathway Metabolism in Treatment-Resistant Depression and Suicidality: A Systematic Review”. CNS Neurol Disord Drug Targets. 2017;16(4):440-453.
- One last remark about the interesting paragraphs 5 or 6:
I think they could be enriched by the observation that, often, it is neurological diseases that are "the ideal territory" for bridging existential considerations and neuroscience. You can refer to these works: “When Sick Brain and Hopelessness Meet: Some Aspects of Suicidality in the Neurological Patient”. CNS Neurol Disord Drug Targets. 2020;19(4):257-263. doi: 10.2174/1871527319666200611130804 (full free PDF text in PubMed) and “Neurological diseases and suicide: from neurobiology to hopelessness]. Rev Med Suisse. 2015 Feb 11;11(461):402-5. (full free PDF text in Google Scholar).
Thank you very much,
Best regards
Author Response
Reviewer 1:
Comments and Suggestions for Authors
Thank you for the opportunity to review this extremely interesting text. It addresses in an original way an issue of increasingly emerging interest and helps to fill the need for an idispensable dialogue between psychotherapy and neuroscience. It is also very well presented with informative and graphically pleasing diagrams.
We sincerely appreciate your valuable time and comments. We are really pleased to hear that the manuscript is able to convey our message.
I would like to highlight just a few points, which the authors are encouraged to address:
- In the abstract (lines 32-33) and in the initial part of the introduction (lines 47-48): I believe it is appropriate to soften the expressions describing psychotherapy as a "potent alternative to the currently prevailing medication-based approaches" and "a preferable otion for intervention of psychiatric disorders." Surely the psychotherapy is a very powerful mean, but I fear that the sentences in object could be generalized and misunderstood. In order not to go into the specific guidelines that, depending on the different disorders, distinguish the indications for psychotherapy versus psychopharmacological treatments, I recommend an expression suggesting "a synergistic action in the vast majority of cases" or a similar expression.
We also realized the issue. The statements have been subdued as follows:
“the existential phenomenological psychotherapy potently plays a synergistic role with the currently prevailing medication-based approaches for the treatment of mood and anxiety disorders.”
“An increasing number of therapists pay more attention to psychotherapy as an effective adjunct maneuver for intervention of psychiatric disorders.”
- Paragraph 3, lines 207-209. This sentence is clearly crucial in the text. I believe it deserves to be enriched by the mention of one of the most dangerous consequences of depression, suicide (just the mention, a more extensive elaboration is not necessary in the economy and purpose of the text). This, also in terms of existential perspective. In fact, the presence of meaning has been shown to be a powerful protective factor against suicide. I suggest two texts, which the authors can read and cite: 1) “The meaning in life in suicidal patients: The presence and the search for constructs. A systematic review”. Medicina (2019), 55(8), 465 (open access full PDF text); 2) “Suicide prevention from a public health perspective. What makes life meaningful? The opinion of some suicidal patients”. Acta Biomed (2020), 91(3-S), 128-134 (full PDF text available on Google Scholar). In the first text there is a historical introduction to the evolution of the concept of meaning and a review of that concept in light of the phenomenon of suicide. The second is instead a qualitative study, which insists - as the authors rightly reiterate - on the subjective experience of meaning in suicidal patients. Both texts advocate psychotherapeutic treatment focused on meaning, following in the footsteps of V. Frankl's logotherapy.
Thank you for your valuable suggestions. The following sentence has been added with reference to the above-mentioned two papers:
“On the other hand, the presence of meaning has been shown to be an active and potent protective factor against the emergence of suicidal tendencies which are among the most dangerous potential consequences of depression.”
- Paragraph 5. This is also an interesting and well-written paragraph. I agree with the authors who highlight the possible role of neuroinflammation. Again, important scientific evidence has been demonstrated in the context of suicide (look, for example, at this work: “Abnormalities in Kynurenine Pathway Metabolism in Treatment-Resistant Depression and Suicidality: A Systematic Review”. CNS Neurol Disord Drug Targets. 2017;16(4):440-453.
Indeed, it is a valuable suggestion. The statement was added to complement the section as follows:
“A systematic review reported KYN metabolism abnormalities in treatment-resistant depression (TRD) and suicidal behavior, proposing the KYN enzymes as novel targets in TRD and suicidality.”
- One last remark about the interesting paragraphs 5 or 6:
I think they could be enriched by the observation that, often, it is neurological diseases that are "the ideal territory" for bridging existential considerations and neuroscience. You can refer to these works: “When Sick Brain and Hopelessness Meet: Some Aspects of Suicidality in the Neurological Patient”. CNS Neurol Disord Drug Targets. 2020;19(4):257-263. doi: 10.2174/1871527319666200611130804 (full free PDF text in PubMed) and “Neurological diseases and suicide: from neurobiology to hopelessness]. Rev Med Suisse. 2015 Feb 11;11(461):402-5. (full free PDF text in Google Scholar).
Thank you for your valuable suggestions. The following statements were added in the last section:
“Meanwhile, it has been observed that neurologic diseases are commonly associated with psychiatric comorbidities, exposing neurologic patients to a sense of hopelessness and consequently a higher risk of suicidal ideation and suicidal behavior. Thus, neurologic diseases may be the ideal territory for bridging existential psychotherapy and neurological science.”
Thank you very much,
Best regards
Reviewer 2 Report
Thank you for the opportunity to revise the manuscript titled: "Crosstalk between Existential Phenomenological Psychotherapy and Neuroscience in Mood and Anxiety Disorders".
- The article is a review, however, it is not specified which type of review.
- Moreover, no info on which methodology has been used to perform the current review. Please, add a methodology section.
- In the introduction, the aim of the article is not well structured and clearly stated to the readers.
- What is the added value of this review? No considerations are provided on this aspect.
- Several parts of the text need reference(s) as for instance lines 57-59, 70-76, 368-375, 381-391, and many many others.
- Discussion and conclusion are completely missing.
- An extensive English revision is needed.
Author Response
Reviewer 2:
Thank you for the opportunity to revise the manuscript titled: "Crosstalk between Existential Phenomenological Psychotherapy and Neuroscience in Mood and Anxiety Disorders".
- The article is a review, however, it is not specified which type of review.
This article is a narrative review with systematic components in Section 4. “Narrative review” was added in Abstract. The Section 4 was revised as follows: “Systematic search was conducted in PubMed/Medline with keyword “meaning-centered” and “psychotherapy” on 16 December 2020. 53 articles were retrieved, and twelve articles were eventually deemed for synthesis.”
- Moreover, no info on which methodology has been used to perform the current review. Please, add a methodology section.
The methods are described in the Section 4 as above.
- In the introduction, the aim of the article is not well structured and clearly stated to the readers.
Thank you for your suggestion. More description was added to Introduction and was structured as follows: background -> psychotherapy -> neurological science -> aims of the article. The sections of the manuscript were rearranged accordingly.
- What is the added value of this review? No considerations are provided on this aspect.
The rationale of this manuscript is presented in the Abstract, Introduction, and the last section. The manuscript was prepared to draw more attention of the readers to translation and integration of philosophy-based psychotherapy and laboratory and clinical-based neuroscience. We noticed that very few articles deal with the converging field and thus, we believe the approach will be a potentially powerful maneuver for the treatment of mood and anxiety disorders.
- Several parts of the text need reference(s) as for instance lines 57-59, 70-76, 368-375, 381-391, and many many others.
The following references were added:
lines 57-59: [10],
lines 70-76: [12],
lines 368-375: [92],
lines 381-391: [105-112].
And more references were added to other parts of the manuscript.
- Discussion and conclusion are completely missing.
The title of the Section 6 was revised as Future Perspective: Bridging the Expertise and the section is intended to conclude the manuscript.
- An extensive English revision is needed.
We corrected the manuscript at our best.
Reviewer 3 Report
In sum, this article demonstrates the development of the Existential phenomenological psychotherapy, the therapeutic methodology, evidence-based accounts of its curative techniques, current understanding of mood and anxiety disorders in neuroscience, and a possible converging path to translate and integrate meaning-centered psychotherapy and neuroscience. Authors conclude that the existential phenomenological approach in psychotherapy is a viable and potent alternative to the currently prevailing medication-based approaches.
The authors may find as follows my main comments/suggestions.
- Talking about mood disorders and anxiety disorders in general is too vague. Authors should better specify what they are referring to in line with the DSM-5 diagnostic criteria
- Authors should also specify the effectiveness of the therapeutical approaches described depending on the severity of mood and anxiety symptoms (mild, moderate, severe)
- Authors cannot conclude that “the existential phenomenological approach in psychotherapy is a viable and potent alternative to the currently prevailing medication-based approaches” without reporting updated scientific literature that investigate the effectiveness comparing psychological and pharmacologic interventions (i.e. Cuijpers P, van Straten A, van Oppen P, Andersson G. Are psychological and pharmacologic interventions equally effective in the treatment of adult depressive disorders? A meta-analysis of comparative studies. J Clin Psychiatry. 2008 Nov;69(11):1675-85; quiz 1839-41).
- A proper paragraph comparing psychological interventions with pharmacological ones is needed.
- Authors should underline the limitations of the clinical trials reported (small sample size, study design, no comparison with pharmacological treatments, …) that affect the achieved results
Author Response
Reviewer 3:
Comments and Suggestions for Authors
In sum, this article demonstrates the development of the Existential phenomenological psychotherapy, the therapeutic methodology, evidence-based accounts of its curative techniques, current understanding of mood and anxiety disorders in neuroscience, and a possible converging path to translate and integrate meaning-centered psychotherapy and neuroscience. Authors conclude that the existential phenomenological approach in psychotherapy is a viable and potent alternative to the currently prevailing medication-based approaches.
The authors may find as follows my main comments/suggestions.
Talking about mood disorders and anxiety disorders in general is too vague. Authors should better specify what they are referring to in line with the DSM-5 diagnostic criteria
Thank you for your valuable suggestion. The first sentence of Introduction was revised as follows:
“Mood and anxiety disorders are one of the most common diagnoses categorized in Diagnostic and Statistical Manual of Mental Disorders, 5th Edition (DSM-5) and the 11th revision of the International Statistical Classification of Diseases and Related Health Problems (ICD-11).”
Authors should also specify the effectiveness of the therapeutical approaches described depending on the severity of mood and anxiety symptoms (mild, moderate, severe)
Additional description was added regarding pharmacotherapy in Introduction.
“A meta-analysis reported that a selective serotonin-reuptake inhibitor (SSRI) fluoxetine was the most effective for the acute treatment of moderate-to-severe depressive disorder in children and adolescents. However, only 40-60% of individuals relieved their symptoms with antidepressant within six to eight weeks.”
Authors cannot conclude that “the existential phenomenological approach in psychotherapy is a viable and potent alternative to the currently prevailing medication-based approaches” without reporting updated scientific literature that investigate the effectiveness comparing psychological and pharmacologic interventions (i.e. Cuijpers P, van Straten A, van Oppen P, Andersson G. Are psychological and pharmacologic interventions equally effective in the treatment of adult depressive disorders? A meta-analysis of comparative studies. J Clin Psychiatry. 2008 Nov;69(11):1675-85; quiz 1839-41).
A proper paragraph comparing psychological interventions with pharmacological ones is needed.
Abstract was revised. Indeed, Cuijpers et al. reported in 2008 that selective serotonin reuptake inhibitors (SSRIs) were more effective than psychological treatments in for the treatment of major depressive disorder. However, Cuijpers et al. in 2020 revealed that combined pharmacotherapy and psychotherapy was superior to monotherapy, while no significant difference was found between pharmacotherapy and psychotherapy in the treatment of depression. The suggested reference was included, and Introduction was revised accordingly.
Authors should underline the limitations of the clinical trials reported (small sample size, study design, no comparison with pharmacological treatments, …) that affect the achieved results
Thank you for your valuable suggestions. The following paragraph was added to the section 4:
“Most MCP studies mentioned above dealt with terminal cancer patients presenting mood symptoms. MCP analysis targeting a population of neurologic or psychiatric diseases, MCP for individuals without comorbidity, and comparison with patients under pharmacotherapy will further reveal the efficacy, the applicability, and the limit of MCP for the treatment of a wide range of diseases.”
Reviewer 4 Report
In this review article the Authors were trying to describe a crosstalk between existential phenomenological psychotherapy and neuroscience in mood and anxiety disorders. The chosen topic is interesting, however, the proportions of article’s text concerning these issues are not properly kept.
My criticism is as follows:
- This article is mainly a psychiatric / psychological treatise with reference to anatomy and neuroimaging and it is difficult to see the achievements in the field of neuroscience there.
- Apart from a few general sentences about the definition of neuroscience and a cursory mention of the kinurenic acid pathway, the authors did not describe neuroscience research in the context of psychiatric diseases. Moreover, while reading this article, one can get the impression that the only neurotransmission system in the central nervous system involved in psychiatric disorders is the glutamate system. And that of course is unjustified narrowing. Thus, considering also the role of other neurotransmitters' systems would improve the manuscript.
- Because of the above I would suggest the Authors to change the title of their article into ”Crosstalk between Existential Phenomenological Psychotherapy and BRAIN CIRCUITS in Mood and Anxiety Disorders” or ”Crosstalk between Existential Phenomenological Psychotherapy and NUROIMAGING in Mood and Anxiety Disorders” (instead of using there the word “Neuroscience”) .
- Also, the subtitle number 5 (Neuroscience of Mood Disorders) should be changed, e.g., to “Neuroimaging of Mood Disorders”.
Author Response
Reviewer 4:
In this review article the Authors were trying to describe a crosstalk between existential phenomenological psychotherapy and neuroscience in mood and anxiety disorders. The chosen topic is interesting, however, the proportions of article’s text concerning these issues are not properly kept.
My criticism is as follows:
- This article is mainly a psychiatric / psychological treatise with reference to anatomy and neuroimaging and it is difficult to see the achievements in the field of neuroscience there.
Thank you for your valuable comment. Most of articles presented in the manuscript deal with abnormal functions and diseases. So, it turns out more suitable to use the term “neurological science”. Neuroscience was replaced with neurological science.
- Apart from a few general sentences about the definition of neuroscience and a cursory mention of the kinurenic acid pathway, the authors did not describe neuroscience research in the context of psychiatric diseases. Moreover, while reading this article, one can get the impression that the only neurotransmission system in the central nervous system involved in psychiatric disorders is the glutamate system. And that of course is unjustified narrowing. Thus, considering also the role of other neurotransmitters' systems would improve the manuscript.
To avoid the confusion, the following description was added.
“Several neurotransmitters including serotonin, dopamine, adrenaline, histamine, gamma-aminobutyric acid, and peptides play an important role in the pathogenesis of mood and anxiety disorders. Glutamatergic neural transmission draws increasing attention because normal human brain functions …”
- Because of the above I would suggest the Authors to change the title of their article into ”Crosstalk between Existential Phenomenological Psychotherapy and BRAIN CIRCUITS in Mood and Anxiety Disorders” or ”Crosstalk between Existential Phenomenological Psychotherapy and NUROIMAGING in Mood and Anxiety Disorders” (instead of using there the word “Neuroscience”) .
Thank you for your valuable suggestions. We intended to present a short description of the recent advance. As the Reviewer suggested the the manuscript does not cover neuroscience. But we would like to add a little bit more than brain circuits or neuroimaging. Neurological science fits better and the title was revised accordingly.
- Also, the subtitle number 5 (Neuroscience of Mood Disorders) should be changed, e.g., to “Neuroimaging of Mood Disorders”.
The title of the section was revised as suggested.
Reviewer 5 Report
In the Review, the authors discuss the critical problem of the polarization of psychotherapy and neuroscience, which seems to be an important issue to find a more effective therapeutic approach for mood and anxiety disorders. The article message, however, should be provided clearer, specifically in the abstract and conclusions sections. In the current version, it is unclear whether the article's main goal is to convince readers to decrease polarization between psychotherapy and neuroscience or increase polarization for psychotherapy. Moreover, the authors should reorganize the manuscript to clarify and convince the readers that the manuscript talks about "crosstalk" between psychotherapy and neuroscience in mood and anxiety disorders.
Specific comments:
- In the abstract (verse 33), the authors conclude that EPP can be an alternative (not kind of support) to pharmacotherapy. Please explain clearly the aim/the main message of the review article.
- Please decrease the number of keywords in the manuscript. Three to ten pertinent keywords are allowed in one Article in Biomedicines journal. See instruction: https://www.mdpi.com/journal/biomedicines/instructions#preparation
- Verse 57-59: The introduction about studying pathogenesis and new drug development using in vitro and in vivo models is not explained further in the body text. Please elaborate more precisely benefits of these studies in the body text of the Review Article. Otherwise, delete an indicated sentence from the Introduction section and explain why "Neuroscience" story is narrowed to fMRI results description.
- The main message could be clearer by reorganizing the information inside of the manuscript. Therefore, my recommendation is to give, between the introduction and conclusions sections, two main chapters with names, e.g. in the first: "Existential Psychotherapy's Approach to Treating Mood and Anxiety Disorders" and in the second: "Neuroscientific Approach to Treating Mood and Anxiety Disorders." Each of them could be divided into sub-chapters for a) general description of each approach; b) providing scientific knowledge about the mechanisms of mood and anxiety disorders resulting from research using a psychotherapeutic and neuroscientific approach, respectively c) effectiveness of treatment by usage each of approach.
- In the “neuroscientific” approach description (in the body text), authors should add pharmacotherapy possibilities and other ways for treating mood and anxiety disorders based on neuroscience knowledge and appropriate literature.
- In the conclusion section: "Bridging the Expertise," authors should clearly state and elaborate where is the possibility for the integration of the psychotherapy and neuroscience knowledge to understand the better mechanism of mood and anxiety disorders and improve their treatment in the future ( or describe precisely limitations of such cooperation).
- Chapter 2 "Existential Phenomenological Psychotherapy" is lengthy, contains information not related directly to Article subject and should be deleted from the manuscript. I recommend to shorten significantly physiological background of Phenomenology and move necessary information to the introduction section. It is likely, a decrease in the number of Figures will be necessary. To undoubtedly valuable information on the details of Phenomenology's history and assumptions, the reader can be referred to the selected literature cited in the manuscript.
- The relationship between psychotherapy and neuroscience approaches and depression as a marker for neurodegenerative and other disorders is not clear in the manuscript. Please, describe more precisely this issue or remove this information from the body text (verse401-413 and Figure 7; verse 432-434).
Minor comments:
- Verse 40-41: Please add a reference to the number of people suffering from mood and anxiety disorders.
- Please correct typos in the manuscript: I found e.g. "per cent" (verse 49); double "in in" (verse 57); "depress" (verse 392).
- Please provide a more precise column title for Table 1. Current names are confusing and not self-described. The reader is not able to predict differences of content in columns "Techniques" and "Methods" because the current name of columns: "techniques" and "methods" are and they are words usually used interchangeably (synonyms).
- For each verse of Table 1, please add a literature reference.
Author Response
Reviewer 5:
In the Review, the authors discuss the critical problem of the polarization of psychotherapy and neuroscience, which seems to be an important issue to find a more effective therapeutic approach for mood and anxiety disorders. The article message, however, should be provided clearer, specifically in the abstract and conclusions sections. In the current version, it is unclear whether the article's main goal is to convince readers to decrease polarization between psychotherapy and neuroscience or increase polarization for psychotherapy. Moreover, the authors should reorganize the manuscript to clarify and convince the readers that the manuscript talks about "crosstalk" between psychotherapy and neuroscience in mood and anxiety disorders.
Specific comments:
- In the abstract (verse 33), the authors conclude that EPP can be an alternative (not kind of support) to pharmacotherapy. Please explain clearly the aim/the main message of the review article.
Thank you for your valuable comment. The passage was revised accordingly.
“…, concluding that the existential phenomenological psychotherapy potently plays a synergistic role with the currently prevailing medication-based approaches for the treatment of mood and anxiety disorders.”
- Please decrease the number of keywords in the manuscript. Three to ten pertinent keywords are allowed in one Article in Biomedicines journal. See instruction: https://www.mdpi.com/journal/biomedicines/instructions#preparation
The number of keywords was reduced accordingly.
- Verse 57-59: The introduction about studying pathogenesis and new drug development using in vitro and in vivo models is not explained further in the body text. Please elaborate more precisely benefits of these studies in the body text of the Review Article. Otherwise, delete an indicated sentence from the Introduction section and explain why "Neuroscience" story is narrowed to fMRI results description.
Thank you for your suggestions. The term neuroscience was replaced with more specific areas of research such as neuroimaging research and neurological studies. Furthermore, a reference was added.
- The main message could be clearer by reorganizing the information inside of the manuscript. Therefore, my recommendation is to give, between the introduction and conclusions sections, two main chapters with names, e.g. in the first: "Existential Psychotherapy's Approach to Treating Mood and Anxiety Disorders" and in the second: "Neuroscientific Approach to Treating Mood and Anxiety Disorders." Each of them could be divided into sub-chapters for a) general description of each approach; b) providing scientific knowledge about the mechanisms of mood and anxiety disorders resulting from research using a psychotherapeutic and neuroscientific approach, respectively c) effectiveness of treatment by usage each of approach.
The heads of sections were modified and rearranged accordingly.
- In the “neuroscientific” approach description (in the body text), authors should add pharmacotherapy possibilities and other ways for treating mood and anxiety disorders based on neuroscience knowledge and appropriate literature.
More descriptions on neuroscientific approaches were added as follows:
“Several neurotransmitters including serotonin, dopamine, adrenaline, histamine, gamma-aminobutyric acid, and peptides play an important role in the pathogenesis of mood and anxiety disorders. SSRIs, selective serotonin and norepinephrine re-uptake inhibitors (SNRIs) and the monoamine oxidase inhibitors (MAOIs) are major classes of antidepressants currently prescribed for the treatment of depression and anxiety. SSRIs, SNRIs, and MAORs all acts on components of neurotransmission. Glutamatergic neural transmission draws increasing attention because normal human brain functions …”
- In the conclusion section: "Bridging the Expertise," authors should clearly state and elaborate where is the possibility for the integration of the psychotherapy and neuroscience knowledge to understand the better mechanism of mood and anxiety disorders and improve their treatment in the future ( or describe precisely limitations of such cooperation).
The following paragraph and reference were added to present possible approach to bridge two expertise:
“Meanwhile, it has been observed that neurologic diseases are commonly associated with psychiatric comorbidities, exposing neurologic patients to a sense of hopelessness and consequently a higher risk of suicidal ideation and suicidal behavior. Thus, neurologic diseases may be the ideal territory for bridging existential psychotherapy and neurological science.”
- Chapter 2 "Existential Phenomenological Psychotherapy" is lengthy, contains information not related directly to Article subject and should be deleted from the manuscript. I recommend to shorten significantly physiological background of Phenomenology and move necessary information to the introduction section. It is likely, a decrease in the number of Figures will be necessary. To undoubtedly valuable information on the details of Phenomenology's history and assumptions, the reader can be referred to the selected literature cited in the manuscript.
Thank you for the suggestions. The section describes essence of meaning-based psychotherapy, especially for clinicians and neuroscientists who are not familiar with meaning-based psychotherapy. Figures 2-5 are rearranged to make the part more concise and informative.
- The relationship between psychotherapy and neuroscience approaches and depression as a marker for neurodegenerative and other disorders is not clear in the manuscript. Please, describe more precisely this issue or remove this information from the body text (verse401-413 and Figure 7; verse 432-434).
They were rephrased and additional description was added to the caption of the figure as follows:
“ The serum plasma profiles may serve as a potential predictive biomarker for the choice of antidepressants. A food or fortified food product beneficial to physiological body functions, …”
“Psychological stress causes depression, anxiety, and cognitive impairment. Depression is a measurable indicator which …”
“… a desirable interventional target …”
Minor comments:
- Verse 40-41: Please add a reference to the number of people suffering from mood and anxiety disorders.
The reference was added.
- Please correct typos in the manuscript: I found e.g. "per cent" (verse 49); double "in in" (verse 57); "depress" (verse 392).
They are corrected accordingly.
- Please provide a more precise column title for Table 1. Current names are confusing and not self-described. The reader is not able to predict differences of content in columns "Techniques" and "Methods" because the current name of columns: "techniques" and "methods" are and they are words usually used interchangeably (synonyms).
The Table is modified accordingly.
- For each verse of Table 1, please add a literature reference.
The reference was added accordingly.
Round 2
Reviewer 1 Report
The authors extensively reviewed the manuscript and responded thoroughly and accurately to the reviewers' comments.
Thank you,
Best regards
Author Response
Reviewer 1:
The authors extensively reviewed the manuscript and responded thoroughly and accurately to the reviewers' comments.
Thank you,
Best regards
Response: we all sincerely appreciate your valuable time and comments for the manuscript.
Reviewer 2 Report
Thank you for having taken into account my previous suggestions. However, the manuscript still misses the theoretical background that informs and sustains the review, the methods section used, and the importance of the results.
Moreover, it is not clear when the results of the review start compared to the introduction or authors' thought.
Still, the conclusions are missing. What can you suggest based on your results? what is the main aspect highlighted by the literature search?
Lastly, a narrative review is something different compared to a systematic review. If it is a systematic review, as stated in the text, then the Authors should follow the PRISMA guidelines.
Please, in the next rebuttal letter for each point clearly specify in which page and lines you made the changes.
Author Response
Reviewer 2:
Thank you for having taken into account my previous suggestions. However, the manuscript still misses the theoretical background that informs and sustains the review, the methods section used, and the importance of the results.
The manuscript consists of the formation of meaning-centered psychotherapy, the clinical evidence of its effectiveness, the neurological science of depression and anxiety, and finally future perspective. They are presented in order. The method and the result are relevant to and described in the section 2.2. ( Page 9, Lines 265-267, Page 10, Lines 298-300)
Moreover, it is not clear when the results of the review start compared to the introduction or authors' thought.
Response: This review article is a narrative review of perspective presenting the current issues and future perspective to inspire collaboration of philosophy-based psychotherapy and neuroscience. The results are only relevant to the section of systematic review to present the evidence of the effectiveness of meaning-centered psychotherapy.
Page 10, Lines 298-300: “In summary, both IMCP and MCGP are more effective than supportive psychotherapy, counselling, or supportive care. MCGP nurtures personal growth and positive relations. MCGP is effective in the long term and more cost-effective. However, …”
Still, the conclusions are missing. What can you suggest based on your results? what is the main aspect highlighted by the literature search?
Response: The conclusion is presented as a summary in the end of the section 2.2. and presented only in the subsection, as the perspective of this narrative review is presented in the end of the narrative review.
As above.
Page 15, Line 465.
Lastly, a narrative review is something different compared to a systematic review. If it is a systematic review, as stated in the text, then the Authors should follow the PRISMA guidelines.
Response: The PRISMA flow chart is provided in Supplement 1.
Page 25.
Please, in the next rebuttal letter for each point clearly specify in which page and lines you made the changes.
The pages and lines are indicated for each correction.
Response: We sincerely appreciate your critical reading and valuable comments.
Reviewer 4 Report
The authors have been able to correctly revise my previous comments and the manuscript has been improved. However, I have one minor comment to the revised version of the manuscript, as follows:
The Authors erased the word “neuroscience” in the entire manuscript and replaced everywhere in the text with the “neurological science”. This was done automatically and in several lines the previous term should be reinstated, i.e., lines 69, 75, 473,
Author Response
Reviewer 4:
The authors have been able to correctly revise my previous comments and the manuscript has been improved. However, I have one minor comment to the revised version of the manuscript, as follows:
The Authors erased the word “neuroscience” in the entire manuscript and replaced everywhere in the text with the “neurological science”. This was done automatically and in several lines the previous term should be reinstated, i.e., lines 69, 75, 473,
Response: Thank you for your valuable suggestions. The manuscript was corrected accordingly.
Reviewer 5 Report
The authors addressed most of my critical remarks and applied necessary corrections into the current version of the Article, which significantly improved its scientific quality. I have only one minor comment:
Please correct explanation for SNRI abbreviation: should be >selective norepinephrine reuptake inhibitor<
Author Response
Reviewer 5:
The authors addressed most of my critical remarks and applied necessary corrections into the current version of the Article, which significantly improved its scientific quality. I have only one minor comment:
Please correct explanation for SNRI abbreviation: should be >selective norepinephrine reuptake inhibitor<
Response: Thank you for your kind attention. The abbreviation was corrected accordingly.